

# Chemistry, taxonomy and ecology of the potentially chimpanzee-dispersed *Vepris teva* sp.nov. (Rutaceae) endangered in coastal thicket in the Congo Republic

Moses K. Langat[1], Teva Kami[2] and Martin Cheek[1]

[1] Science, Royal Botanic Gardens, Kew, Richmond, United Kingdom
[2] Herbier National, Institut de Recherche National en Sciences Exactes et Naturelles (IRSEN), Brazzaville, Republic of Congo

## ABSTRACT

Continuing a survey of the chemistry of species of the largely continental African genus *Vepris*, we investigate a species previously referred to as *Vepris* sp. 1 of Congo. From the leaves of *Vepris* sp. 1 we report six compounds. The compounds were three furoquinoline alkaloids, kokusaginine (1), maculine (2), and flindersiamine (3), two acridone alkaloids, arborinine (4) and 1-hydroxy-3-methoxy-10-methylacridone (5), and the triterpenoid, ß-amyrin (6). Compounds 1–4 are commonly isolated from other *Vepris* species, compound **5** has been reported before once, from Malagasy *Vepris pilosa*, while this is the first report of ß-amyrin from *Vepris*. This combination of compounds has never before been reported from any species of *Vepris*. We test the hypothesis that *Vepris* sp. 1 is new to science and formally describe it as *Vepris teva*, unique in the genus in that the trifoliolate leaves are subsessile, with the median petiolule far exceeding the petiole in length. Similar fleshy-leathery four-locular syncarpous fruits are otherwise only known in the genus in *Vepris glaberrima* (formerly the monotypic genus *Oriciopsis* Engl.), a potential sister species, but requiring further investigation to confirm this phylogenetic position. We briefly characterise the unusual and poorly documented Atlantic coast equatorial ecosystem, where *Vepris teva* is restricted to evergreen thicket on white sand, unusual in a genus usually confined to evergreen forest. This endemic-rich ecosystem with a unique amphibian as well as plants, extends along the coastline from the mouth of the Congo River to southern Rio Muni, a distance of about 1,000 km, traversing five countries. We map and illustrate *Vepris teva* and assess its extinction risk as Endangered (EN B1ab(iii)+B2ab(iii)) using the *IUCN, 2012* standard. Only three locations are known, and threats include port and oil refinery construction and associated activities, with only one protected location, the Jane Goodall Institute's Tchimpounga Reserve. Initial evidence indicates that the seeds of *Vepris teva* are dispersed by chimpanzees, previously unreported in the genus.

Corresponding author
Martin Cheek, m.cheek@kew.org

## INTRODUCTION

As part of a series of studies of the chemistry of *Vepris* led by the first author, material from a Congolese *Vepris* previously referred to in field studies as *Vepris sp*. 1 of Pointe Noire was investigated. In this article we present the chemical results, compare the taxon morphologically within *Vepris*, test the hypothesis that this taxon is new to science and formally name it as *Vepris teva* Cheek. We also present data on its ecology in coastal thicket on white sand, a poorly known and highly threatened ecosystem on the equatorial Atlantic coast of Africa. Initial field observations suggest that the seeds of *Vepris teva* are dispersed by chimpanzees.

*Vepris* Comm. ex A. Juss. (Rutaceae-Toddalieae), is a genus with 91 accepted species, 23 in Madagascar and the Comores and 68 in Continental Africa with one species extending to Arabia and another endemic to India (*POWO, continuously updated*). The genus was last revised for tropical Africa by *Verdoorn (1926)*. Founded on the Flore du Cameroun account of *Letouzey (1963a)*, nine new species were recently described from Cameroon (*Onana & Chevillotte, 2015*; *Cheek, Gosline & Onana, 2018a*; *Onana, Cheek & Chevillotte, 2019*; *Cheek & Onana, 2021*; *Cheek, Hatt & Onana, 2021*), taking the total in Cameroon to 25 species, the highest number for any country globally. The greatest concentration of *Vepris* species in Cameroon is within the Cross-Sanaga Interval (*Cheek et al., 2001*) with 15 species of *Vepris* of which nine are endemic to the Interval. The Cross-Sanaga has the highest species and generic diversity per degree square in tropical Africa (*Barthlott, Lauer & Placke, 1996*; *Dagallier et al., 2020*) including endemic genera such as *Medusandra* Brenan (Peridiscaceae, *Breteler, Bakker & Jongkind, 2015*; *Soltis et al., 2007*).

By comparison, neighbouring Gabon has just eight species, seven species of *Vepris* listed in the Gabon checklist (*Sosef et al., 2006*) and *V. lecomteana*. (Pierre) Cheek & T. Heller. Meanwhile, just one species, *Vepris lecomteana* is listed for Congo-Brazzaville (*POWO, continuously updated*), illustrating how under-recorded the Flora of this biodiverse country is. However, it is almost certain that additional species for this country are represented in the Paris herbarium, P., but we have not been able to access this during the pandemic. Several Cameroon species are threatened (*Onana & Cheek, 2011*) and in one case considered globally extinct (*Cheek, Gosline & Onana, 2018a*), although only two currently appear on the IUCN Red List: *Vepris lecomteana* (Pierre) Cheek & T. Heller (Vulnerable, *Cheek, 2004*) and *Vepris trifoliolata* (Eng.) Mziray (Vulnerable, *World Conservation Monitoring Centre, 1998*). In other parts of Africa species are even more highly threatened, *e.g.*, the Critically Endangered *Vepris laurifolia* (Hutch. & Dalziel) O. Lachenaud & Onana of Guinea-Ivory Coast (formerly *V. felicis* Breteler, *Cheek, 2017*; *Lachenaud & Onana, 2021*).

In continental Africa, *Vepris* are easily recognised. They differ from all other Rutaceae because they have digitately (1–)3(–5)-foliolate (not pinnate) leaves, and unarmed (not spiny) stems. The genus consists of evergreen shrubs and trees, predominantly of tropical lowland evergreen forest, but with some species extending into submontane forests and some into drier forests and woodland. *Vepris* species are often indicators of good quality,

relatively undisturbed evergreen forest since they are not pioneers. New species are steadily coming to light (*Cheek et al., 2019*).

Species of *Vepris* in Africa extend from South Africa, *e.g., Vepris natalensis* (Sond.) Mziray, to the Guinean woodland in the fringes of the Sahara Desert (*Vepris heterophylla* (Engl.) Letouzey). *Mziray (1992)* subsumed the genera *Araliopsis* Engl., *Diphasia* Pierre, *Diphasiopsis* Mendonça, *Oricia* Pierre, *Oriciopsis* Engl., *Teclea* Delile, and *Toddaliopsis* Engl. into *Vepris*, although several species were only formally transferred subsequently (*e.g.,* *Harris, 2000*; *Gereau, 2001*; *Cheek, Oben & Heller, 2009*; *Onana & Chevillotte, 2015*). Mziray's conclusions were largely confirmed by the molecular phylogenetic studies of *Morton (2017)* but Morton's sampling was limited, identifications appeared problematic (several species appear simultaneously in different parts of the phylogenetic trees) and more molecular work would be desirable. Morton studied about 14 taxa of *Vepris*, all from eastern Africa. More recently *Appelhans & Wen (2020)* focussing on Rutaceae of Madagascar have found that the genus *Ivodea* Capuron is sister to *Vepris* and that a Malagasy *Vepris* is sister to those of Africa. However, the vast majority of the African species including all those of West and Congolian Africa, remain unsampled leaving the possibility open of changes to the topology of the phylogenetic tree when this is addressed.

Characteristics of some of the formerly recognised genera are useful today in grouping species. The "araliopsoid" species have firm, subglobose, four-locular fruit syncarpous with four external grooves; the "oriciopsoid" soft, fleshy four-locular syncarpous fruit; "oricioid" species are four-locular and apocarpous in fruit; the fruits of "diphasioid" species are laterally compressed in one plane, bilocular and bilobed at the apex; while "tecleoid" species are unilocular in fruit and one-seeded, lacking external lobes or grooves. There is limited support for these groupings in Morton's study,

Due to the essential oils distributed in their leaves, and the alkaloids and terpenoids distributed in their roots, bark and leaves, several species of *Vepris* have traditional medicinal value (*Burkill, 1997*). Burkill details the uses, essential oils and alkaloids known from five species in west Africa: *Vepris hiernii* Gereau (as *Diphasia klaineana* Pierre), *Vepris suaveolens* (Engl.) Mziray (as *Teclea suaveolens* Engl.), *Vepris afzelii* (Engl.) Mziray (as *Teclea afzelii* Engl.), *Vepris heterophylla* (Engl.) Letouzey (as *Teclea sudanica* A. Chev.) and *Vepris verdoorniana* (Exell & Mendonça) Mziray (as *Teclea verdoorniana* Exell & Mendonça) (*Burkill, 1997*: 651–653). Research into the characterisation and anti-microbial and anti-malarial applications of alkaloid and limonoid compounds in *Vepris* is active and ongoing (*e.g., Atangana et al., 2017*), although sometimes published under generic names no longer in current use, *e.g., Wansi et al. (2008)*. Applications include as synergists for insecticides (*Langat, 2011*). *Cheplogoi et al. (2008)* and *Imbenzi et al. (2014)* respectively list 14 and 15 species of *Vepris* that have been studied for such compounds. A review of ethnomedicinal uses, phytochemistry, and pharmacology of the genus *Vepris* was recently published by *Ombito, Chi & Wansi (2021)*, listing 213 different secondary compounds, mainly alkaloids and furo- and pyroquinolines, isolated from 32 species of the genus, although the identification of several of the species listed needs checking. However, few of these compounds have been screened for any of their potential applications. Recently, *Langat et al. (2021)* have published three new acridones and reported multi-layered

synergistic anti-microbial activity from *Vepris gossweileri* (I.Verd.) Mziray, recently renamed as *Vepris africana* (Hook.f ex Benth.) Lachenaud & Onana (*Lachenaud & Onana, 2021*).

## MATERIALS AND METHODS

### Taxonomy

*The electronic version of this article in Portable Document Format (PDF) will represent a published work according to the International Code of Nomenclature for algae, fungi, and plants (ICN), and hence the new names contained in the electronic version are effectively published under that Code from the electronic edition alone. In addition, new names contained in this work which have been issued with identifiers by IPNI will eventually be made available to the Global Names Index. The IPNI LSIDs can be resolved and the associated information viewed through any standard web browser by appending the LSID contained in this publication to the prefix "http://ipni.org/". The online version of this work is archived and available from the following digital repositories: PeerJ, PubMed Central, and CLOCKSS.*

Fieldwork in the Republic of Congo resulting in the specimens and observations cited in this article was conducted with the collaboration and support of the CERVE (Centre d'Etudes sur les Ressources Végétales) (currently named IRSEN (Institut de Recherche National en Sciences Exactes et Naturelles, Brazzaville))-National Herbarium of Congo and Royal Botanic Gardens, Kew beginning in 2010 under research permit (Autorisation de Recherche) 021/MRS/DGRST/DMAST (issued 2 November 2010), and the specimens were exported under permit (Autorisation D'Exportation des Énchantillons Botaniques) number 003/CERVE/57/2011 (issued 4 December 2011). At the Royal Botanic Gardens, Kew, fieldwork was approved by the Institutional Review Board of Kew entitled the Overseas Fieldwork Committee (OFC) for which the registration number was OFC 490-3–490-6 (2010–2011). The most complete set of duplicates for all specimens made was deposited at IEC, the remainder exported to K for identification and distribution following standard practice.

The taxonomic study is based on herbarium specimens and observations of live material in Congo-Brazzaville made by the second two authors and their colleagues in 2010–2012. Herbarium citations follow Index Herbariorum (*Thiers, continuously updated*), nomenclature follows *Turland et al. (2018)* and binomial authorities follow *IPNI (2021)*. The methodology for the surveys in which the specimens were collected is given in *Cheek & Cable (1997)*. All specimens cited have been seen. Material of the suspected new species was compared morphologically with material of all other species African *Vepris*, principally at K, but also using material and images from BM, EA, BR, FHO, G, GC, HNG, P and YA. Specimens at WAG were viewed on the Naturalis website (https://bioportal.naturalis.nl/). The main online herbarium used during the study apart from that of WAG was that of P (https://science.mnhn.fr/all/search). Herbarium material was examined with a Leica Wild M8 dissecting binocular microscope fitted with an eyepiece graticule measuring in units of 0.025 mm at maximum magnification. The drawing was made with the same equipment using a Leica 308700 camera lucida attachment.

## Ecology

Once *Vepris teva* was detected as being new to science, seemingly restricted to coastal white sands (with other new and threatened species) in July 2011 we conducted desk-top studies using Google Earth to find other areas of this poorly studied habitat where it survives in the Republic of Congo. We visited these areas in November–December 2012, travelling by road from the northern border with Gabon to the southern border with Angola (Cabinda) to visit targeted sites, but also ground-truthing sites in different but adjoining habitats to test whether the species was indeed restricted to the white sand habitat.

## Chemistry

Samples were made from live plants (preferable to preserved material for chemical analysis) cultivated from seed at the Royal Botanic Gardens, Kew which were associated with the herbarium collections *Mpandzou et al.* 1754 (IEC, K) collected with the authorisation of Herbier National, Institut de Recherche National en Sciences Exactes et Naturelles (IRSEN formerly CERVE, see permit references below), Cité Scientifique de Brazzaville. Two plants, grown from seed, were cultivated in 3 inch (c. 8 cm) diam. plastic pots, using a free-draining potting medium of 25% Coir (coconut husk fibre), 17% each of Seramis (baked expanded clay granules), medium Perlite (amorphous volcanic glass) and fine bark, 12% loam and 12% grit, with a slow-release fertiliser. Plants were cultivated in a glasshouse in full light, with temperature maintained in the range 21–25 deg. Celsius. The humidity goal is 70%, achieved by spraying over and damping down the floor at least twice a day. Watering is usually once per day, and feeding twice a week in summer, once in winter, with an additional weekly Kelp feed.

The plants grew very slowly, reaching only 15 cm in height, and not becoming reproductive, even after 10 years of cultivation, before they died in 2021. The cause of death is unknown. This suggests that there is scope to develop an improved cultivation protocol. The RBG, Kew reference number for this accession was 2019-14.

Spectroscopic and spectrometric analysis were conducted as follows: the FTIR spectra were recorded using a Perkin-Elmer Frontier/Spotlight 200 spectrometer, and the acquired data used to determine the functional groups present in the compounds. 1D and 2D NMR spectra were recorded in $CDCl_3$ on a 400 MHz Bruker AVANCE NMR instrument at room temperature. Chemical shifts ($\delta$) are expressed in ppm and were referenced against the solvent resonances at $\delta_H$ 7.26 and $\delta_C$ 77.23 ppm for $^1$H and $^{13}$C NMR for $CDCl_3$. Accurate masses, for determination of molecular formulae of the compounds, were recorded on a Thermo Scientific Orbitrap Fusion spectrometer. Purity of compounds was monitored *via* thin layer chromatography (TLC) using pre-coated aluminium-backed plates (silica gel 60 $F_{254}$; Merck, Kenilworth, NJ, USA) and compounds were visualised by UV radiation at 254 nm and then using an anisaldehyde spray reagent (1% *p*-anisaldehyde:2% $H_2SO_4$:97% cold MeOH) followed by heating. Final purifications used preparative thin layer chromatography (Merck 818133) and gravity column chromatography that was carried out using a 2 cm diameter column, which were packed with silica gel (Merck Art. 9385) in selected solvent systems.

The leaves were freeze-dried and ground to fine powder using a blender. The dried leaves (37 g) were successively extracted using methylene chloride ($CH_2Cl_2$) solvent to target non-polar and semi-polar compounds and methanol solvent (MeOH) to target polar compounds. The $CH_2Cl_2$ and MeOH extracts obtained were 1.7 g and 4.3 g respectively. The methylene chloride extract was subjected to gravity column chromatography packed with a 1:1 blend of silica gel merck 9385 and eluted isocratically using 10% ethyl acetate in methylene chloride, collecting 35 mL. The fractions were monitored using TLC and fractions with the same retention times were pooled. Fractions 12–13 gave compound **5** which was determined to be 1-hydroxy-3-methoxy-10-methylacridone (*Haensel & Cybulski, 1978*). Fractions 14–18 gave compound **6** which was determined to be ß-amyrin (*Okoye et al., 2014*). Fractions 23–25 gave compound **2** which was determined to be maculine (*Vaquette et al., 1976*). Fractions 34–35 gave compound **4** which was determined to be arborinine (*Haensel & Cybulski, 1978*). Fractions 40–47 gave compound **3** which was determined to be flindersiamine (*Vaquette et al., 1976*) and fractions 55–63 gave compound **1** which was determined to be kokusaginine (*Pusset et al., 1991*).

## Extinction risk assessment

Points were georeferenced using locality information from herbarium specimens. The map was made using simplemappr (*Shorthouse, 2010*). The conservation assessment was made using the categories and criteria of *IUCN (2012)*, EOO was calculated with GeoCat (*Bachman et al., 2011*). Threats were observed by the second two authors directly in the field in Republic of Congo.

# RESULTS

## Chemistry

The structures of the alkaloids were determined based on comprehensive spectroscopic and spectrometric analysis, and the spectra of the known compounds were compared to those previously reported. The compounds were three furoquinoline alkaloids, kokusaginine (1) (*Pusset et al., 1991*), maculine (2) (*Vaquette et al., 1976*) and flindersiamine (3) (*Vaquette et al., 1976*), two acridone alkaloids, arborinine (4) (*Haensel & Cybulski, 1978*) and 1-hydroxy-3-methoxy-10-methylacridone (5) (*Haensel & Cybulski, 1978*), and one triterpenoid, ß-amyrin (6) (*Okoye et al., 2014*) (Fig. 1).

This combination of secondary compounds matches none of those reported from the 32 taxa of *Vepris* that have been chemically investigated to date (*Ombito, Chi & Wansi, 2021*). While compounds 1–4 are relatively widespread in the investigated species, each having been recorded in 9–12 taxa, compound 5 is recorded only in one species, *Vepris pilosa* (Baker)I.Verd. (now a synonym of *V. glomerata* (F.Hoffm.)Engl.). However, compounds 2 & 3 have not been recorded in *Vepris glomerata*. Compound 6 has not been reported previously from *Vepris*. The chemical results therefore do not conflict with the morphologically-based conclusion that *Vepris* sp. 1 is new to science (see below).

**Figure 1 Furoquinoline and acridone alkaloids isolated from *Vepris teva*.** Furoquinoline alkaloids: kokusaginine (1), maculine (2), and flindersiamine (3). Acridone alkaloids: arborinine (4) and 1-hydroxy-3-methoxy-10-methylacridone (5).

## Morphology

The morphological characteristics of *Vepris* sp. 1 are highly unusual within the genus. The subsessile leaves of the fertile stems with the median petiolule far exceeding the petiolule in length is unique within the genus. The leathery-walled, syncarpous, four-loculed, and slightly lobed, subverrucate fruits which have fleshy-juicy mesocarp surrounding the seeds are otherwise known in only one species, *Vepris glaberrima*, formerly segregated as the monotypic genus *Oriciopsis*. The two species share several unusual characters, such as numerous parallel, straight secondary and intersecondary nerves, with few quaternary nerves, sparse oil glands and long median petiolules, suggesting they are sister species. *Vepris* sp. 1 however differs greatly from the last species in the characters indicated below in Table 1 and in the diagnosis below.

***Vepris teva*** Cheek sp. nov.–Figs. 2 and 3.
*Type.* Republic of Congo, Kouilou, Port 8, along coast just northwest of Pointe Noire, near Pointe Indienne, 4° 42′ 46.5′S, 11° 49′ 18.3″E, 10 m elev., fl. 23 Nov1 mber 2011, *T. Kami et al.* 1227 (holotype K000875074; isotypes EA, IEC, MO, P, US)

Syn. *Vepris* sp. 1 of Pointe Noire.

Diagnosis: differs from all known trifoliolate species of *Vepris* in the median petiolule far exceeding the petiole in length (usually by a factor of 2–4 times) on reproductive stems, especially near the stem apex (*vs* petiole exceeding petiolule in length in all other species). Most similar to *Vepris glaberrima* (Engl.) J.B.Hall ex D.J. Harris in the soft, leathery-fleshy, four-loculed, and slightly lobed, subverrucate syncarpous fruit, differing in the inflorescences exceeding the petiole in length (*vs* shorter than the petioles), the secondary

**Table 1 Morphological characters separating *Vepris teva* from *Vepris glaberrima*.**

| Characters | *Vepris teva* | *Vepris glaberrima* |
|---|---|---|
| Petiole length (cm) | 0.14–0.4(–0.75) | 4–10 |
| Petiolule length (cm) | (0.3–)0.5–1.1(–1.6) | 1.1–2.7 |
| Median leaflet texture and dimensions (cm) | Coriaceous (4.5–)5.5–10.6(–14.3) × (2–)2.7–3.5(–5.2) | Papyraceous 7.5–18(–20) × 2.5–8.3 |
| Acumen length (mm) | (3–)4–9(–10) | 13–19(–20) |
| Oil gland dots on abaxial surface | Raised above surface, moderately conspicuous | Not visible |
| Petal length (mm) | 3.3–3.5 | 4.8–5.1 |
| Calyx lobe | Well-developed 0.4–0.7 mm long | Not developed |
| Fruit shape | Subglobose | Ovoid |
| Fruit size (mm) | 11–14 × 11–13 | 20–25 × 16–20 |

nerves (10–)11–16(–18) on each side of the midrib (*vs* 20–30), the leaflet apices shortly rounded-acuminate (*vs* long, acutely acuminate) and other characters shown in Table 1 above.

*Dioecious (probably) shrub* 0.5–1.5 m tall, moderately branched, glabrous apart from the bracteole margins. Stems terete, internodes (0–)0.9–5.3(–7.5) cm long, (1.5–)2–3.5 (–4) mm diam. at lowest leafy node, epidermis glossy, medium brown, finely longitudinally striate, at length splitting longitudinally, soon lenticellate, lenticels very sparse white, orbicular or elliptic, often longitudinally divided in two, (0.3–)0.5–1.3(–2.25) × (0.2–) 0.3–0.6(–0.9) mm.

   *Leaves* alternate, trifoliolate (rarely unifoliolate), *median leaflet* usually slightly longer than lateral leaflets, elliptic, less usually obovate-elliptic, (4.5–)5.5–10.6(–14.3) × (2–) 2.7–3.5(–5.2) cm, acumen broadly triangular (0.3–)0.4–0.9(–1) × 0.25–0.45 cm, apex rounded, base acute-decurrent, secondary nerves (10–)11–16(–18) on each side of the midrib, arising at 50–80° from the midrib, straight, united by a slightly looping inframarginal vein 0.6–1.2 mm from the margin (Figs. 1A–1C), tertiary nerves conspicuous on the abaxial but not adaxial surface, mainly parallel to the secondary nerves, uniting transversely only in the outer part of the blade (Fig. 1B), quaternary nerves inconspicuous; oil gland dots translucent in transmitted light, inconspicuous on the adaxial surface, conspicuous but concolorous, raised on the abaxial surface (Figs. 1B and 1C), (2–)4–7(–8) per mm$^2$, the diameter of the glands 0.1–0.225 mm; lateral leaflets as the median leaflet, but (4.7–)5.5–8.7(–12.7) × (1.9–)2.2–3.4(–4.25) cm, base acute. Petiolules canaliculate, those of fertile stems exceeding the petiole in length (Figs. 1A and 1C), (0.3–) 0.5–1.1(–1.6) cm long, those of median leaflets much longer than the lateral; articulated at junction with the petiole. Petiole canaliculate, those of fertile stems 0.15–0.4(–0.75) cm long, those of sterile stems (*e.g., Mpandzou* 1653, IEC, K) much longer, 4–5.2 cm long. *Inflorescences* terminal, less usually axillary in the most distal subapical node(s), thyrsoid, contracted, about as wide as long 0.4–0.85 cm diam., 5–20-flowered, main axis with 1–3 pairs of 1–3-flowered cymes ± evenly spaced from the base; bracts quadrate-triangular, 0.5 × 0.8 mm. Pedicels each subtended by a second order bract and two bracteoles, all

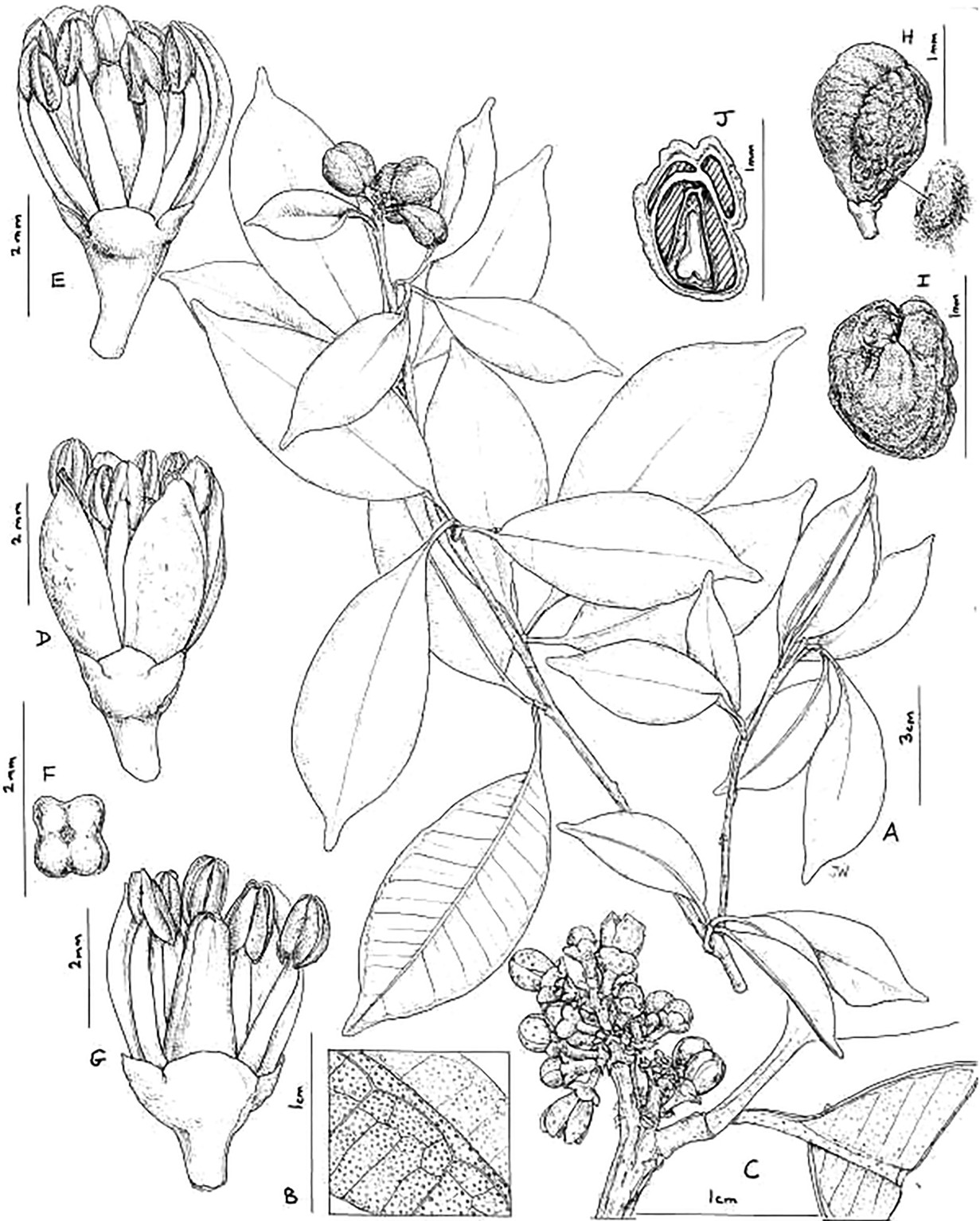

**Figure 2** *Vepris teva.* (A) Habit, fruiting stem; (B) detail of leaflet blade showing oil gland dots and nervation, lower surface of blade in foreground; (C) detail of male inflorescence; (D) male flower, side view; (E) as (D), two petals removed to show stamens; (F) pistil of male flower, four-lobed, viewed from above; (G) male flower, two petals and three stamens removed to show pistil; (H) mature fruit, side view (seed on right); (I) fruit, plan view; (J) fruit, transverse section, showing four locules, three aborted and one with seed. (A and H–J) from *Mpandzou et al.*, 1198, (B and G) from *Kami, T. et al* 1356; (C–F) from *Kami, T. et al.*, 1227. Drawn by Juliet Williamson, CC-BY-NC-ND.

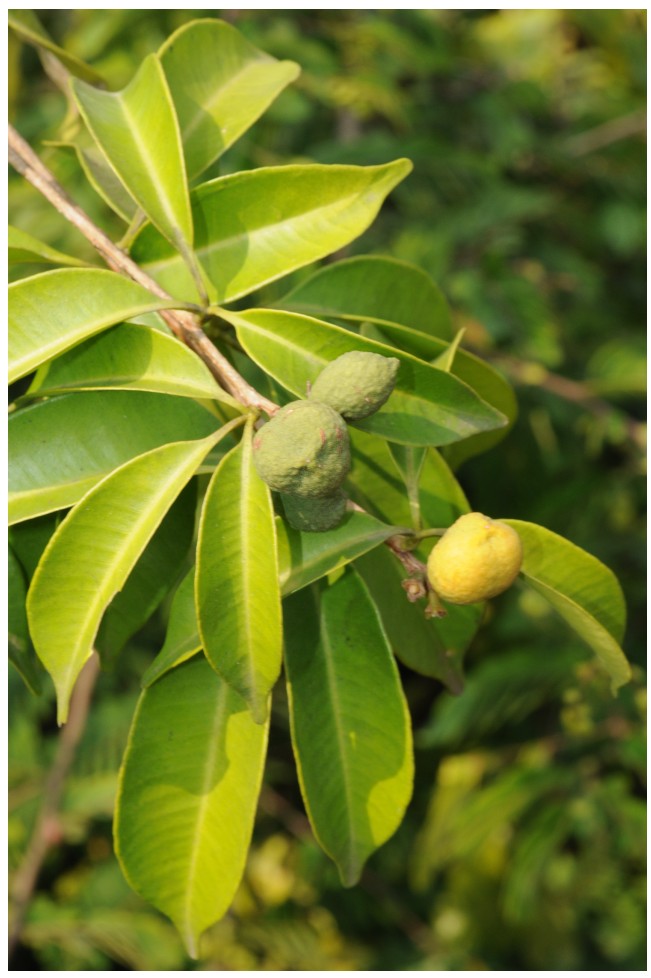

**Figure 3** ***Vepris teva.*** Shrub with mature (yellow) and immature (green) fruits. Note the sessile leaves. From *Mpandzou et al.*, 1198 (IEC, K). Photo by M. Cheek.

ovate-triangular, 0.5–0.6 × 0.3 mm, margins sparingly minutely simple hairy. *Female flowers* unknown, apart from parts persisting in fruit. Stigma discoid, peltate, subsessile, 1 mm diam. *Male flowers* with pedicel 1.2–1.5 × 0.5 mm, terete, widening to 0.8–1.2 mm wide below the sepals, lacking conspicuous glands.

*Calyx* with sepals 4, imbricate, erect, broader than long, transversely semi-elliptic, 0.4–0.7(–0.9) × 1.1–1.6 mm, apex broadly rounded or rounded-obtuse. *Petals* 4, oblanceolate, concave, erect, 3.2–3.5 × 1.4–1.5 mm, apex obtuse, rounded or minutely retuse, margin membranous, central part thickened with numerous raised oil-glands conspicuous on the abaxial surface. *Stamens* 8, erect, free, slightly exceeding petals, subequal, the outer 4 with slightly longer filaments than the inner 4, filaments terete, 2.5–3.0 mm long, 0.4 mm wide at base, narrowing gradually to 0.2 mm wide at apex; anthers submedifixed, introrse to lateral dehiscence, oblong-ovate, 1–1.2 × 0.6 mm; disc inconspicuous; ovary (pistillode) obclavate-4-angled/fluted, 3–3.1 × 1.2–1.3 mm, 0.7 mm wide at apex, apex rounded with a slight central depression, in plan view 4-lobed (Fig. 1F) stigmas punctate, minute; 4-locules each biovulate. *Fruit* in terminal clusters of (1–)2–4,

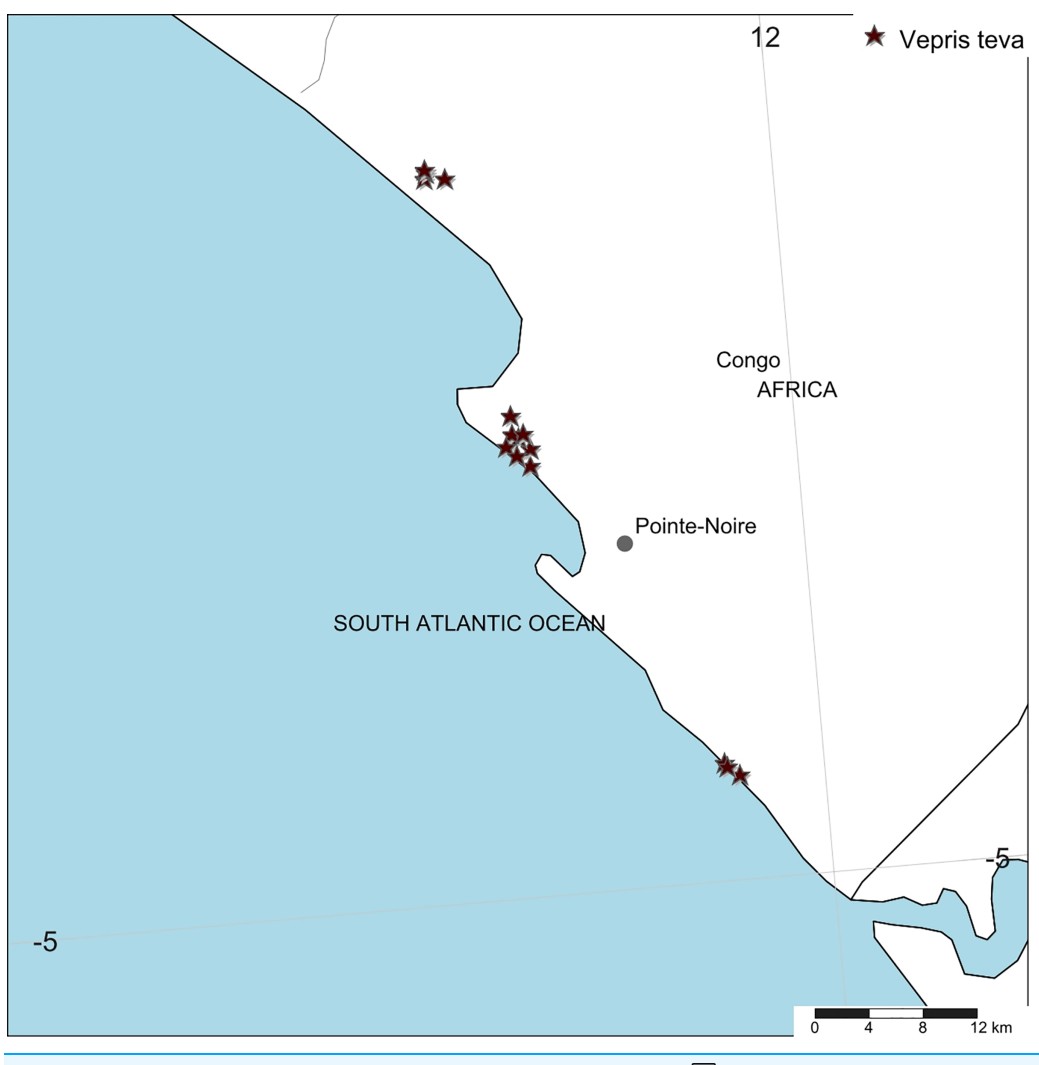

**Figure 4** *Vepris teva*. Global distribution. 

yellow-orange, subglobose or obovoid, apex rounded-slightly depressed, longitudinally 4-grooved to slightly lobed, 11–14 × 11–13 mm, 4-locular but usually with 1–2 locules incompletely formed, 1–2(–4)-seeded by abortion. Pericarp leathery, surface minutely wrinkled to slightly verrucate, 0.5–0.7 mm thick, endocarp bony 0.05 mm thick; mesocarp liquid, sweet to the taste. *Seed* encased in a cartilaginous, vascular endocarp, ellipsoid, 4–4.5 × 2.4 × 1.8–2 mm (Fig. 1H).

**Distribution.** Republic of Congo, Kouilou Department, Tchimpounga to Djeno Fig. 4.

**Ecology. The evergreen thicket ecosystem on white sand on the Atlantic coast of Africa**
*Vepris teva* is restricted to the key vegetation type, thicket on white sand, that occurs in a coastal ecosystem extending along the equatorial Atlantic coast of Africa in the southern hemisphere. This ecosystem extends from the mouth of the Congo River in the Democratic Republic of Congo, along the coast through Angolan Cabinda, Republic of Congo into southern Gabon, with some areas thought to extend into Rio Muni of Equatorial Guinea.

It extends discontinuously along the coastline for about 1,000 km and extends inland for between 100–3,000 m, and is based on old, partly dissected and flattened, highly leached white sand dunes that run parallel to the coast, alternating with lower, wet or seasonally wet drainage areas often developing black peaty soils, that can develop either swamp forest *e.g.*, with *Alstonia congensis* Engl. trees and an understorey of *Acrostichum aureum* L. or a wetland grassland community including Cyperaceae, *Utricularia*, *Drosera*, Xyridaceae and *Stipularia africana* P.Beauv. Thicket develops on the top of the ridges. In the ecotone transition areas, the shallow slopes between the thicket and wetland area, a sparse grassland develops on the upper part of the white sand with *Chlorophytum*, *Dissotis congolensis* (Cogn. ex Büttner) Jacq.-Fél., *Eulophia caricifolia* (Rchb.f.) Summerh., and in the lower damper, seasonally inundated parts of the ecotone, grass species including *Anadelphia hamata* Stapf, and herbs such as *Neurotheca corymbosa* Hua, both globally restricted to this vegetation type.

On the seaward side of the dunes a sea-shore community including halophytes develops with succulent species such as *Sansevieria longiflora* Sims, while in brackish inlets the Atlantic mangrove community is formed, including *Rhizophora racemosa* G.Mey. On the landward side of the ecosystem two substrates interface with the white sand, red-brown loam, and grey sand which each have their own communities of grassland species. This ecosystem has rarely been referred to in the literature. The best account is probably that by *Vande Weghe (2007)* regarding the areas in southern Gabon. He refers to a sand-burrowing toad *Hemisus perretii* Laurent (*Vande Weghe, 2007*: 250) that is restricted to this ecosystem. *Cheek, van der Burgt & Pickering (2010)* have characterised this ecosystem in the Republic of Congo after our studies in 2010–2012, recognising the nine habitats referred to above, six of which have threatened, or provisionally threatened plant species often restricted to this ecosystem. However, Critically Endangered (CR) and Endangered (EN) species are reported only from two of the habitats: the thicket (13 species) and sparse grassland on white sand habitats (two species). The evergreen thicket which is the habitat to which *Vepris teva* is restricted is only 2–3 m tall and consists of shrubs, intermixed with very few emergent tree species *e.g., Hyphaene guineensis* Schumach. & Thonn. and *Tessmannia dawei* J.Léonard, numerous climbers and several herbs, characteristically:

Common shrubs: *Chrysobalanus icaco* subsp. *icaco* L., *Syzygium guineense* var. *guineense (*Willd.) D C., *Ochna multiflora* DC, *Dalbergia grandibracteata* De Wild., *Manilkara lacera* (Baker) Dubard, *Premna serratifolia* L., *Rytigynia dewevrei* Robyns, *Trichoscypha imbricata* Engl., *Tricalysia coriacea* (Benth.) Hiern, *Vismia affinis* Oliv., *Psychotria kimuenzae* De Wild., *Psydrax moandensis* Bridson, *Thomandersia butayei* De Wild., *Baphia leptostemma* subsp. *leptostemma* Baill. and *Leptactina mannii* Hook.f.,

Common climbers: *Pentarhopalopilia marquesii* (Engl.) Hiepko, *Uvaria versicolor* Pierre ex Engl. & Diels, *Calycobolus cabrae* (De Wild. & T.Durand) Heine, *Jasminum kwangense* Liben and *Ancylobothrys scandens* (Schumach. & Thonn.) Pichon

Common herbs: *Pseuderanthemum lindavianum* De Wild. & T. Durand, *Coleus calaminthoides* Baker and *Dracaena braunii* Engl.

The grassland is maintained by dry season fires which kill most or all woody plants (*Vande Weghe, 2007*), none of which, curiously, appear fire-adapted. Such grassland fires were recorded by us in Congo in July 2010. Owing to the sparsity of the herbs in the grassland, fires are not intense due to the low fuel-load. It is sometimes possible for shrubs to establish in grassland. *Chrysobalanus icaco* is a shrub which, once established, in the absence of fire, can spread laterally to form a thicket in which other species of shrub and tree can colonise, leading towards succession from grassland to thicket.

Among the endemic and highly threatened shrub species of this ecosystem, several apart from *Vepris teva*, were found to be new to science. Of these, those recently published are *Dracaena marina* Damen (*Damen et al., 2018*), *Baphia vili* Cheek (*Cheek, Kami & Kami, 2014*), *Salacia arenicola* Gosline (*Gosline, Cheek & Kami, 2014*). However, several additional species remain to be published.

**Local names and uses.** None are recorded.

**Etymology.** Named for Teva Kami, lead collector of the type specimen, who played a key role in the discovery of this species and further research upon it concerning interactions with chimpanzees (see below).

**Conservation.** *Vepris teva* is known from nine specimens and seven sight-records made between July 2011–February 2012. These equate to an area of occupation of 40 km$^2$ using the IUCN-preferred 4 km$^2$ gridcells, and an extent of occurrence of 172 km$^2$. It is restricted to coastal thicket on white sand habitat in Republic of Congo. This habitat is thought to extend along the Atlantic coast from coastal DRC to southern Equatorial Guinea. Despite targetted surveys in this habitat through most of this range especially in Congo, *Vepris teva* has only been found at three of the more than eight locations studied. These are: (1) Pointe Noire at Point Indienne; (2) Djeno; (3) Tchimpounga. At the first location, the plants are threatened by an extension northward of the port of Pointe Noire to accommodate export of rock phosphate, manganese, iron-ore for which there is currently insufficient port capacity. Plants of *Vepris teva* are also threatened by cutting of their coastal thicket habitat for charcoal and for clearance for housing. At the second location, plants are threatened by the infrastructure and activities of the Total E&P Congo petro-chemical plant (part of TotalEnergie S.A.), the installation of which appears to have destroyed much of the habitat of *Vepris teva* at this location. Indirectly, Total, as the major employer at Djeno attracting labour which requires local accommodation, appears to have stimulated an expansion of urbanisation, resulting in *Vepris teva* habitat being parcelled for sale as house building plots (M Cheek, 2012, personal observations). At Tchimpounga, a reserve created by the Jane Goodall Foundation to protect chimpanzees, *Vepris teva* uniquely appears protected and secure so long as this venture is supported. In view of the EOO and AOO, and the threats stated, we here assess *Vepris teva* as Endangered (EN B1ab(iii)+B2ab(iii)).

## Additional specimens and observations

Republic of Congo, Dept. Kouilou, Tchimpounga, along coast NW of Pointe Noire, about 7 km SE of the Kouilou River Bridge, in the Tchimpounga Chimpanzee Sanctuary, fl. 3 December 2011, *T. Kami et al.*,1356 (IEC, K000875073); Tchimpounga pointe 2 zone ex gorillon, fl. 5 November 2012, *Mpandzou et al.*,1641 (IEC, K000875078); Tchimpounga, bas-Kouilou, Bois de Singe, st. 6 December 2012; *Mpandzou et al.*,1754 (IEC, K000875081); Tchimpounga, Point 1 zone de soleil, fl. 2 November 2012, *T. Kami et al.*,1421 (BR, IEC, K000875083); Tchimpounga, Point 1 zone de soleil, fl. 3 November 2012, *T. Kami* 1437 et al., (IEC, K000875082); Pointe Noire, Port 8, along coast just northwest of Pointe Noire, near Pointe Indienne, 4° 42′ 46.5″S, 11° 49′ 18.3″E, 10 m elev., fl. 23 November 2011, *T. Kami et al.*,1227 (holotype K000875074; isotypes EA, IEC, MO, P, US); Pointe Noire, fr. 9 July 2011, *Mpandzou et al.*,1198 (IEC, K000875072, P); near Pointe Indienne, 24 November 2011, Port Observations 36; ibid. 24 November 2011, Port Observations 61; ibid, 25 November 2011, Port Observations 91; ibid. 26 November 2011, Port Observations 128; 27 November 2011, Port Observations 215; Djeno a côté de Terminal de Djeno, fr. 6 December 2012, *Mpandzou et al.*,1768 (IEC, K000875077); ibid., fr. 6 December 2012, *Mpandzou et al.*,1754 (IEC, K000875081); Djeno, near Total refinery, 15 February 2012, Port Observations 309; ibid., 13 July 2011, Port Observations 303.

**Notes.** We first detected *Vepris teva* in July 2011, during the dry season environmental impact studies for a proposed new port facility near the major existing port of Pointe Noire, Congo's principal port and commercial centre. It was provisionally named as '*Vepris* sp. 1 of Pointe Noire,' and considered likely to be new to science since it matched no other known species of the genus in tropical Africa (*Cheek, van der Burgt & Pickering, 2010*). Further collections were made in a wet season survey in November and December 2011. These surveys resulted in the discovery of many other rare species, usually previously unknown to science, mainly confined to evergreen thicket on coastal white sand. Consequently, surveys in November–December 2012 were made along the length of the Congolese coastline to map the full national extent of this ecosystem and of its rare and threatened species. It was discovered that unlike many of the coastal thicket species, *Vepris teva* is restricted to only a small length of the coast, from Djeno in the south to Tchimpounga in the north. At these three sites, wherever coastal thicket on white sand appears, *Vepris teva* can be fairly frequent. However, when thicket become degraded or damaged, it is no longer present. We believe that our surveys have detected all sites where this species survives in the Republic of Congo. However, it is possible that additional individuals might survive in fragments of original vegetation that were overlooked by us *e. g.*, within the greater area of the city of Pointe Noire, but we consider this unlikely. Also, it is possible that the species might extend to Angola (Cabinda) or southern Gabon which we were not able to visit during the studies referred to in this article. *Vepris teva* sometimes grows alongside another species of the genus, *Vepris africana* (formerly *Vepris gossweileri*). The second species extends into Angola and to S.Tomé (*Lachenaud & Onana, 2021*). The two species cannot be confused since the first is trifoliolate, while the second is

unifoliolate. No *Vepris* species other than these two have been found in coastal thicket on white sand in Congo.

Our knowledge of the phenology of *Vepris teva* is restricted to two periods of the year – July (dry season) and November–December (wet season). At both seasons plants were found in both flower and fruit. All flowers collected are monomorphic and have well-developed stamens with pollen, and are therefore supposed to be male. This is supported by our observation that while the fruits (the undoubted products of female flowers) have a 1 mm diam. peltate, discoid stigma, peltate stigmas have not been in any of the flowers which are therefore male. Since *Vepris* species have been found to be dioecious wherever their reproductive strategies have been studied, we conclude that female flowers remain to be discovered, although female plants (in fruit) have been collected. There is a small possibility that *Vepris teva*, atypically in the genus, might have hermaphroditic flowers.

*Vepris teva* grows very slowly from seed. Three plants cultivated at the Royal Botanic Gardens, Kew from seed collected in 2011, reached only 15–25 cm tall after 5 years. By March 2020, after 8.5 years, the plants had reached 30–45 cm tall, but none had shown signs of flowering. All three of these plants showed the long petiole of sterile plants, and not the characteristic extremely short petioles of reproductive individuals. Thus, the age of sexual maturity is likely to be at least 10 years in *Vepris teva*.

Because *Vepris teva* has a 4-locular syncarpous flower producing a soft fleshy fruit it would formerly have been classified in the monotypic genus *Oriciopsis* Engl. (*Engler, 1931*), together with *V. glaberrima* (*Letouzey, 1963b*: 79–80). The last species occurs in lowland evergreen forest at the junction of Cameroon, C.A.R. and Gabon and is also a small, sparingly branched shrub. It is conceivable that the two are sister species (see above). Four-locular syncarpous ovaries are also seen in the formerly accepted genera *Araliopsis* and *Toddaliopsis*, but in these taxa the fruits are hard. The former occurs in W-C Africa but is a large tree with digitately 5-foliolate leaves, the second a tree of coastal E Africa with a subspiny, "nut-like" fruit.

*Vepris teva* appears unique and is easily recognisable in the genus and in its habitat because of the combination of long petiolules with extremely short petioles on the fertile stems. The petioles can be so contracted that they are almost invisible, and it then appears that there are three simple, leaves, each on its own petiole (in fact leaflets on petiolules) at each node.

**Chimpanzee–mediated seed dispersal.**

During field surveys of the vegetation in which *Vepris teva* occurs, the last author tested the hypothesis that the fruits might conform to the 'spat seed' syndrome characteristic of some primate-dispersed plant species (*Sengupta, McConkey & Radhakrishna, 2015*; *Dominy & Duncan, 2005*). When the base of a fruit plucked from a shrub is sucked, the pleasantly sweet, watery mesocarp with the seed(s) enters the mouth, the juice can be swallowed, and the slippery seed(s) in their endocarp(s) spat out (Fig. 5).

One of the three known locations for *Vepris teva* is the Tchimpounga Reserve, co-managed by the Jane Goodall Institute (*Jane Goodall Institute, 2018*) which focuses on

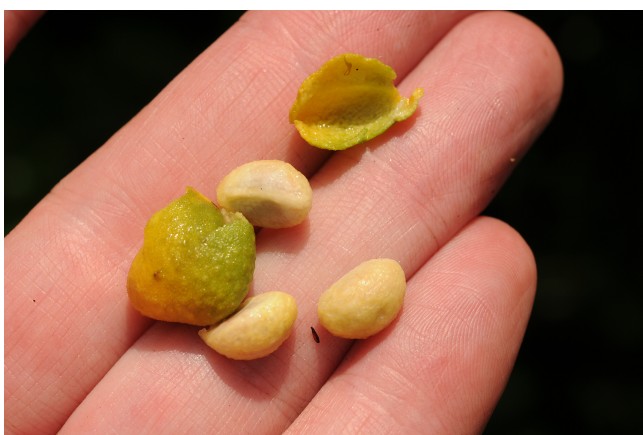

**Figure 5 Vepris teva.** The empty, leathery pericarp after juice abstraction and three spat seeds. Photo: M. Cheek.

chimpanzee conservation and research of both rescued animals from captivity and hunting, and chimpanzees indigenous to the coastal thicket ecosystem (see below). During field observations for doctoral studies of fruit consumption by chimpanzees from tree species by the second author (2014–2020, *Kami, 2021*), chimpanzees were observed to seemingly feed on the fruits of this species of *Vepris*–evidenced by empty pericarps and separated, spat seeds of this species after they had been in the vicinity of the fruiting shrubs. Chimpanzee seed dispersal of *Vepris teva* seems very likely but more detailed studies are needed to confirm this.

## CONCLUSIONS

The description of a known compound, ß-amyrin, for the first time in *Vepris*, is consistent with previous research wherein nearly each species of the genus that is newly chemically investigated contributes additional compounds, often those that are new to science (*e.g.*, *Langat et al., 2021*). The number of species of *Vepris* that have been chemically investigated is rising steadily, numbers having doubled in the last 7 years, with 15 species of *Vepris* reported as investigated by *Imbenzi et al. (2014)* and 32 by *Ombito, Chi & Wansi (2021)*, the last reporting 213 secondary compounds in five classes from *Vepris*. However, this represents only a third of the known species diversity of the genus, which itself is increasing as taxonomists uncover additional species new to science, 12 new species having been published since 2000, a c. 15% increase in numbers for the genus, the recent novelties having been detected mainly in Cameroon (*e.g.*, *Onana & Chevillotte, 2015*). The range of biological activity shown by these compounds is wide and has great potential to address current and future needs of humanity, from medical to agricultural. However, screening for biological activity lags far behind discovery of compounds.

The case of *Vepris teva* illustrates the importance of uncovering and publishing species before they become extinct and become lost forever, together with their potential applications for humanity, in this case secondary compounds with potential *e.g.*, as anti-microbials, natural insecticides and in the case of ß-amyrin, recorded here for the first time in *Vepris*, strong anti-inflammatory action (*Okoye et al., 2014*).

For last 15 years or more, about 2000 new species of flowering plant have been published by scientists each year (*Cheek et al., 2020*), adding to the estimated 369,000 already documented (*Nic Lughadha et al., 2016*). However, only 7.2% of species have been assessed for their threat status and are included on the Red List using the *IUCN (2012)* standard (*Bachman et al., 2019*). Newly discovered species, such as *Vepris teva*, reported in this article, are likely to be threatened, since widespread species tend to have been already discovered. There are notable exceptions to this rule (*e.g.*, *Vepris occidentalis* Cheek & Onana (*Cheek et al., 2019*) a species widespread in West Africa from Guinea to Ghana). However, it is generally the more range-restricted, infrequent species that remain unknown to science. This makes it urgent to find, document and protect such species before they become extinct. Until species are described and known to science, it is difficult to assess them for their IUCN conservation status and so the possibility of protecting them is reduced (*Cheek et al., 2020*). Documented extinctions of plant species are increasing, *e.g.*, in coastal forest of Cameroon, *Oxygyne triandra* Schltr. is considered extinct at its sole locality, the forest at Mabeta-Moliwe in the foothills of Mt Cameroon (*Cheek et al., 2018b*; *Cheek & Williams, 1999*; *Cheek, 1992*), *Inversodicraea bosii* (C.Cusset) Rutish. & Thiv. at the Lobe Falls (*Cheek et al., 2017*) and in Gabon *Pseudohydrosme buettneri* Engl. (*Cheek, Tchiengué & van der Burgt, 2021*) in coastal forest. There are also examples of species that appear to have become extinct even before they are known to science, such as in Cameroon *Vepris bali* Cheek (*Cheek, Gosline & Onana, 2018a*), and in Gabon *Pseudohydrosme bogneri* (*Moxon-Holt & Cheek, 2021*). Human pressures have been the cause of these extinctions in all these cases.

In the Republic of Congo natural habitat is fortunately relatively extensive and intact, but in some specialised ecosystems such as that with the evergreen thicket on white sand habitat described above, to which *Vepris teva* is restricted, large areas have entirely disappeared and others are set to follow them.

Further effort in prioritising high priority areas for plant conservation as Tropical Important Plant Areas (TIPAs), using the revised IPA criteria set out in *Darbyshire et al. (2017)* is being implemented in Guinea (*Couch et al., 2019*). TIPAs are also in progress in countries such as Cameroon, Ethiopia, Mozambique and Uganda and might be extended elsewhere in Africa such as to the Republic of Congo, to reduce the risk of future global extinctions of range-restricted endemic species such as *Vepris teva*.

## ACKNOWLEDGEMENTS

The authors thank Dr Emile Kami, former Head of the National Herbarium of Republic of Congo (IEC, formerly CERVE, now IRSEN) for expediting authorisation for botanical prospection and export of material to RBG, Kew. Our colleagues at CERVE and RBG, Kew, Aydrif Laurel Mpandzou, Raustand Mboungou, Grace Koubemba, Emile Kami, Augustin Ngoliélé, Gilbert Nsongola, Xander van der Burgt, Helen Pickering, Mme Yvette Bongou & Felix Merklinger assisted collecting the material in Congo. Paul Reed, John Merry and Colin Harris of MPD Congo S.A. provided logistic support. At the Jane Goodall Institute's Tchimpounga Reserve, Dr Rebeca Atiencia is thanked for facilitating access for our surveys. Sarah Redstone and colleagues in the Quarantine House RBG, Kew are thanked

for initially growing *Vepris teva* from seed, and we thank especially Bradley Gangadeen of the Science Glass for growing it on under the supervision of Joanna Bates who kindly provided cultivation details. The import and safe use of this wild collected plant material was enabled through the use of Defra Plant Health Licence 2149/194627/5. Jean Michel Onana, National Herbarium of Cameroon is thanked for helpful comments on an earlier version of this article, and for formal reviews we also thank Uday Turaga and an anonymous reviewer, together with Whitney Kistler, editor, for their valuable comments which have improved the article.

### Funding

The authors received no funding for this work.

### Competing Interests

The authors declare that they have no competing interests.

### Author Contributions

- Moses K. Langat conceived and designed the experiments, performed the experiments, analyzed the data, prepared figures and/or tables, authored or reviewed drafts of the article, and approved the final draft.
- Teva Kami conceived and designed the experiments, performed the experiments, analyzed the data, authored or reviewed drafts of the article, field collections & observations, and approved the final draft.
- Martin Cheek conceived and designed the experiments, performed the experiments, analyzed the data, prepared figures and/or tables, authored or reviewed drafts of the article, field observations, and approved the final draft.

### Ethics

The following information was supplied relating to ethical approvals (*i.e.*, approving body and any reference numbers):

At the Royal Botanic Gardens, Kew, fieldwork was approved by the Institutional Review Board of Kew entitled the Overseas Fieldwork Committee (OFC) for which the registration number was OFC 490-3 – 490-6.

### Field Study Permissions

The following information was supplied relating to field study approvals (*i.e.*, approving body and any reference numbers):

Fieldwork in the Republic of Congo resulting in the specimens and observations cited in this article was conducted with the collaboration and support of the CERVE (Centre d'Etudes sur les Ressources Végétales) (currently named IRSEN (Institut de Recherche National en Sciences Exactes et Naturelles, Brazzaville))-National Herbarium of Congo and Royal Botanic Gardens, Kew beginning in 2010 under research permit (Autorisation de Recherche) 021/MRS/DGRST/DMAST (issued 2 Nov 2010).

## Data Availability

NMR spectroscopic data for the compounds isolated from Vepris teva are available in the Supplemental File.

Raw morphological data from the specimens of Vepris teva are available in the Supplemental File.

## New Species Registration

The following information was supplied regarding the registration of a newly described species:

Vepris teva Cheek LSID 77302858-1.

## Supplemental Information

Supplemental information for this article can be found online at http://dx.doi.org/10.7717/peerj.13926#supplemental-information.

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
