# Peer review of "Chemistry, taxonomy and ecology of the potentially chimpanzee-dispersed Vepris teva sp.nov. (Rutaceae) endangered in coastal thicket in the Congo Republic"

_PeerJ, doi:10.7717/peerj.13926_

## Round 0.1 · original submission · Major Revisions

The authors do a good job of describing a novel Vepris species using morphological and biochemical analysis. However, there are areas where they can improve their study. First, the methods section should include the growing conditions of the plant in Royal Botanic Gardens, Kew. The methods can also be written in a more direct manner that reflects the sequence of events in the study. There is also additional information that can be added to the figures. Please see the attached document and reviewers comments for additional areas of improvement.

Are their any plans to add molecular work characterize this species? Molecular characterization of species is very common and does not carry the financial burden it once did.

·

Basic reporting

The manuscript was well written and it made a good read. The authors provided enough background/literature to make sure the reader understands and appreciates the scope and importance of the work. The objectives are clearly mentioned and the data sufficiently validates the findings of the authors.

Experimental design

The experimental design was well explained in the manuscript. Here are a few questions for the authors:
1. Lines 195-200: What made the authors decide upon the extraction protocol that they used in the study? Did they use the extraction protocol targeting the extraction of those compounds mentioned in the same paragraph? A little more detail here helps the prospective audience.
2. Lines 195-197: What exactly did the authors mean by this sentence, " The dried leaves (37g) were successively extracted using methylene chloride and methanol solvents to yield 1.7g and 4.3g respectively."
3. Materials and Methods, Chemistry, Lines 175-180: why did the authors chose to cultivate the plant from seed at the Royal Botanic Gardens instead of trying to collect them at those locations where these plants are usually found? It is totally understandable if the authors justify this by mentioning the difficulty in obtaining the samples from the original locations.

Validity of the findings

The authors clearly explained their findings and the impact of the study from a conservation point of view. All findings by the authors are duly validated by the raw data presented in the study.

Reviewer 2 ·

Basic reporting

In this manuscript, the authors provide a complete description of the species Vepris teva (Rutaceae) based on morphological and chemical evidence. The manuscript is well written, with professional English, detailed background, and information about the species. Below I pose severl questions/comments.

Experimental design

1) The authors used field observation and chemical analytical techniques to describe a new species of Vepris. In terms of experimental design, more information about how and where field collections took place should be described. For example, Figure 4 illustrates the distribution of the new species along the coast. It appears to be highly clumped and restricted to the coast (the authors mention that the species is restricted to white sands), but I do not understand how much of this reported distribution (in figure 4) is due to sampling effort vs habitat preference. I would like the authors to provide more details on how exactly the distribution was determined. Figure 4 could benefit from a satellite image for the baselayer of the map to illustrate how vegetation type changes as one moves away from the coast. The authors could also improve the distribution using some sort of distribution modeling approach or simply by illustrating how widespread they anticipate the species being given its habitat preference.


2) What were the FTIR spectra used for? Can you provide the context of what these data provide and indicate if you compared the spectra between known species and the one described here. Did you present these results in the results section?

3) Line 176. How does taking chemical extracts from cultivated individuals impact the chemical profile? It seems that these species are growing out of their native habitat and may be altered by conditions at the Royal Botanical Gardens. Can you justify this approach or provide references to where this approach has been used?

4) The structure of the methods section is confusing. I would structure the methods in the order of how the samples were processed/collected. fieldwork > Sampling/sample storage > extraction and chemical characterisation. Right now it runs backward where you discuss the instruments you use and then below describe the sampling/storage.


5) On a similar note on lines 200 - 207 you present the main results of chemistry in the methods and then repeat it below. For clarity leave all findings in the results.

Validity of the findings

1) The authors provide detailed chemical information on the new species. I appreciate that they went through the effort to structurally characterize the compounds present in the samples. One question I have is if the authors have a sense of what percent of the entire chemical profile they observed/structurally characterized? It is very often the case that species invest in dozens of compounds, some of which will be low abundance. I know the authors cannot reasonably characterize all lower abundance compounds via NMR, but I would like to know if the compounds they present make up the majority of chemical investment, or if they are a small portion of all compounds produced by this species. On a similar note, the authors should provide context for why they chose a methylene chloride extract, this seems highly targeted chemical approach, as many untargeted chemical studies use an intermediate polarity mixture of Methanol: water as a solvent. Do you think the choice of extraction buffer would change the findings?

2) When the authors discuss the distribution of the new species, can you describe if this species grows in sympatry with other Vepris species? The authors describe how this species can be identified morphologically compared to other known Vepris species, but do other species coexist in the same habitat? Put another way, if a research team were to visit coast the white sand coast of the Congo Republic, would they need to distinguish among multiple Vepris species, or would Vepris teva be the only one present and thus be identified simply because of distribution/location.

---

## Round 0.2 · Minor Revisions

Overall the manuscript does a good job of evaluating the discovery of a new species of Vepris in the Congo Republic. However, there are a few typographical issues and questions that need to be addressed. These can be found in the annotated pdf document from Reviewer 3.

·

Basic reporting

The authors did a commendable job to address the questions and concerns of all reviewers.

Experimental design

The experimental design section is better organized in the revised manuscript. The methods section is explained in greater detail which makes it easier for anyone who wants to use this study for their work.

Validity of the findings

No comments

Additional comments

NONE

·

Basic reporting

Review of "Chemistry, taxonomy and ecology of the potentially chimpanzee-dispersed Vepris teva sp.nov. (Rutaceae) endangered in coastal thicket in the Congo Republic"

Reviewer Carel Jongkind, May 2022


The manuscript is well written and easy to read. Enough data is presented to show that Vepris teva is indeed a new species. It is certainly worth publishing.
More than the usual information is collected on the ecology and distribution of this new species, this makes the Extinction risk assessment detailed and more reliable than in most other cases.
The description, table and nice illustrations give/show more than enough details from the new species.

Having seen the comments from the earlier reviewers and the answers from the authors I think that enough is done to satisfy the reviewers and future readers of the manuscript/publication.

My only question is about the number of Vepris species in Congo-Brazzaville. Did the authors check the Paris herbarium (online) to count the number of species identified from that country? If so please mention this. If not please show more uncertainty about the number of species known from there. I expect that Plants of the World Online is in this case not a reliable source of information because there is no Checklist or Flora from Congo Brazzaville to copy information from. (I get already 3 Vepris species from Congo Brazzaville when I extract data from the BR herbarium online, that kind of extraction is regrettably not possible from at Paris online).

I have no comments on the chemistry part of the manuscript because that is not my expertise.

I have added a few small changes and comments to the PDF text.

Experimental design

Fine

Validity of the findings

Enough data is presented to show that Vepris teva is indeed a new species.
More than the usual information is collected on the ecology and distribution of this new species, this makes the Extinction risk assessment detailed and more reliable than in most other cases.
The description, table and nice illustrations give/show more than enough details from the new species.

I have no comments on the chemistry part of the manuscript because that is not my expertise.

Additional comments

My only question is about the number of Vepris species in Congo-Brazzaville. Did the authors check the Paris herbarium (online) to count the number of species identified from that country? If so please mention this. If not please show more uncertainty about the number of species known from there. I expect that Plants of the World Online is in this case not a reliable source of information because there is no Checklist or Flora from Congo Brazzaville to copy information from. (I get already 3 Vepris species from Congo Brazzaville when I extract data from the BR herbarium online, that kind of extraction is regrettably not possible from at Paris online).

---

## Round 0.3 · accepted · Accept

The authors did an excellent job of addressing the reviewers comments and I am happy to accept this study for publication.